# Methyl Jasmonate and Nanoparticles Doped with Methyl Jasmonate affect the Cell Wall Composition of Monastrell Grape Skins

**DOI:** 10.3390/molecules28031478

**Published:** 2023-02-03

**Authors:** María José Giménez-Bañón, Diego Fernando Paladines-Quezada, Juan Daniel Moreno-Olivares, Juan Antonio Bleda-Sánchez, José Ignacio Fernández-Fernández, Belén Parra-Torrejón, Gloria Belén Ramírez-Rodríguez, José Manuel Delgado-López, Rocío Gil-Muñoz

**Affiliations:** 1Murcian Institute of Agricultural and Environment Research and Development (IMIDA), Ctra. La Alberca s/n, 30150 Murcia, Spain; 2Department of Inorganic Chemistry, Faculty of Science, University of Granada, 18071 Granada, Spain

**Keywords:** nanotechnology, nanoparticle, cellulose, hemicellulose, proteins, uronic acids, phenols

## Abstract

The structural composition of the cell wall of grape skins is related to the cell wall integrity and subsequent extraction of the different compounds that are contained inside vacuoles and also the cell wall breakdown products. Different reports have established that methyl jasmonate (MeJ) produces changes in the composition of the grape skin cell wall. The use of elicitors to promote the production of secondary metabolites in grapes has been studied in several reports; however, its study linked to nanotechnology is less developed. These facts led us to study the effect of methyl jasmonate (MeJ) and nanoparticles doped with MeJ (nano-MeJ) on the cell walls of Monastrell grapes during three seasons. Both treatments tended to increase cell wall material (CWM) and caused changes in different components of the skin cell walls. In 2019 and 2021, proteins were enlarged in both MeJ and nano-MeJ-treated grapes. A general decrease in total phenolic compounds was detected with both treatments, in addition to an increment in uronic acids when the grapes were well ripened. MeJ and nano-MeJ produced a diminution in the amount of cellulose in contrast to an increase in hemicellulose. It should be noted that the effects with nano-MeJ treatment occurred at a dose 10 times lower than with MeJ treatment.

## 1. Introduction

One of the most important factors affecting the quality of red wine is the colour, which is linked to polyphenols and anthocyanins. Anthocyanins and other polyphenols are located inside the vacuoles of the grape skin cell wall and diffuse into the wine during fermentation [1]. The structure and composition of the grape skin cell wall affect the extractability of the compounds it contains [2]. Another factor to consider is that the grape skin cell wall is the main source of pectin polysaccharides in wine [3], which affects the technological and sensory properties of wines, influencing the colloidal stability of wines due to their ability to interact and combine with tannins.

Other factors that impact the extractability of phenolic compounds are grape variety and origin [4], agronomic practices and degree of ripening [5,6]. Different oenological practices focus on breaking down the grape skin to favour the release of polyphenols. The use of enzymes and cold pre-fermentative maceration have established an influence on the composition of cell wall material (CWM) [4,7]. Another issue to consider is that processes of absorption of anthocyanins occur on the cell wall material (CWM) during fermentation [8].

MeJ has been used as an elicitor that causes the production of secondary metabolites such as polyphenols [9,10,11] and aromatic compounds [12,13], and modifies the amino acid profile [14]. The use of this elicitor has been studied as an inducer of the plant defence system [15] and specifically in grapes against *Botrytis cinerea* [16] and downy mildew [17]. In addition, another study showed that MeJ also inhabits the activity of cell wall-degrading enzymes, which delays softening in blueberries [18]. The influence of MeJ on the composition of the Monastrell grape skin cell wall has been studied [19,20], finding that exogenous application causes significant changes in several components of the skin cell walls, and that the response has a varietal and meteorological dependence. Apolinar-Valiente et al. [21] studied the composition of polysaccharides and oligosaccharides that came from the cell wall in Monastrell wines and detected a lower release in wines from grapes treated with MeJ and other elicitors, suggesting an increase in cell wall rigidity that hindered the extraction of the compounds of interest into wine.

The use of nanotechnology, due to slow release, has been shown to improve the efficiency of MeJ in grapes. A study on Monastrell grapes and wines showed an increase in amino acid composition in foliar treatments with MeJ and nano-methyl jasmonate (nano-MeJ) in a similar way [22]. In addition, other researchers showed an increase in stilbene content in Monastrell wines from grapes treated with MeJ and nano-MeJ [23]. Finally, Giménez-Bañón et al. [24] found a higher concentration of fruity ester compounds in Monastrell wines from nano-MeJ-treated grapes.

MeJ is an expensive compound with high volatility, which is why it is necessary to optimize the application method in order to reduce the dose and avoid losses due to volatization or degradation. This would help to reduce costs and achieve a more sustainable agriculture. That is the reason that led us to investigate how grapes and wines are affected after the application of MeJ supported on nanoparticles, using lower doses than treatment with MeJ applied in a conventional way. So far, there has been no report to study the ability of these nano-MeJ to modify the structure and composition of the cell wall; therefore, the aim of this paper was to investigate the composition of the cell wall of Monastrell grapes treated with MeJ and nano-MeJ and its relation to the chromatic and phenolic characteristics obtained in the wines. In addition, because the season is an important factor to consider, the study included its effect, as it was carried out during three consecutive seasons 2019, 2020, and 2021.

## 2. Results and Discussion

### 2.1. Physicochemical Characteristics of the Grapes

Different physicochemical variables of the grapes at harvest were analysed: weight (100 berries), °Brix, titratable acidity (TA) and pH. As can be seen in Table 1, the harvest date was earlier in 2019 than the rest of the vintages studied. The reason for this was the torrential rains that occurred during the second half of September, which made it necessary to bring forward the harvest to avoid the development of any mould attack.

Regarding berry size, an increase was observed only in grapes treated in the last season. Portu et al. [11] found an increase in MeJ-treated Tempranillo grapes during only one season. However, neither Garde-Cerdán et al. [25] nor Pérez-Alvarez et al. [26] showed differences in grapes treated with MeJ or nanoparticles doped with MeJ in Tempranillo or Monastrell grapes, respectively.

The treatments did not affect soluble solids (°Brix) or pH. Pérez-Alvarez et al. [26] likewise found no differences in these variables in Monastrell grapes treated with nanoparticles doped with MeJ in two water regimes—rainfed and deficit irrigation strategy.

As for TA, we can observe that MeJ-treated grapes showed an increase in this variable during the three years of study. Only in 2019 the grapes treated with nano-MeJ showed higher acidity. These results are similar to those found by Garde-Cerdán et al. [25] who reported higher acidity in Tempranillo grapes treated with MeJ and nano-MeJ only during one season. Other authors, such as Paladines-Quezada et al. [19], also detected a different response by TA depending on the season in Monastrell grapes.

### 2.2. Isolation of Cell Wall Material (CWM)

Table 2 shows the percentage of dry skin on grapes and the percentage of isolated CWM in the dry skin. The highest percentage of dry skin appeared in MeJ-treated grapes, being statistically significant in 2019 and 2021. These values could be interesting from an oenological point of view, as grape skins are the major source of colour and aroma compounds [5]. Paladines-Quezada et al. [19,20], in two different studies carried out in 2015 and 2017 on Monastrell grapes, found no differences regarding the amount of skin in control and MeJ-treated grapes. The reason for the differences found in the different studies could be the way of expressing the results: this author showed results referring to fresh skins, and in our study, they are indicated in dry skins to eliminate possible errors induced by different remaining water in the grape skin. On the contrary, when the grapes were treated with nano-MeJ, the grapes showed a tendency towards a decrease in the percentage of skin, as can be seen in the lower values obtained with respect to the control grapes in 2019 and 2020. This lower amount of skin could initially indicate a lower release of phenolic compounds into the wine.

Regarding the percentage of isolated CWM, treated grapes obtained higher values compared to control grapes. In 2019, the highest percentages were shown in MeJ and nano-MeJ-treated grapes, in 2020 it was nano-MeJ-treated grapes, and in the last season, again it was MeJ-treated grapes followed by nano-MeJ-treated grapes. This general increment in CWM in treated grapes could denote a thicker cell wall. Different researchers [21,27] suggest that the application of elicitors in vineyards causes reinforcement of the cell wall, hindering extraction in the winemaking process.

### 2.3. Analysis of Grape Skin Cell Wall Composition

Figure 1 shows the profile of the skin cell wall with respect to season (2019, 2020, and 2021) and treatments (control, MeJ and nano-MeJ). The different components analysed were proteins, total phenolic compounds, uronic acids, cellulosic glucose and non-cellulosic glucose.

#### 2.3.1. Proteins

The cell wall proteome can be divided into nine functional classes, including proteins that act on cell wall carbohydrates (expansins), enzymes (proteases, oxidoreductases), proteins involved in signalling (arabinogalactan proteins), and structural proteins (hydroxyproline-rich glycoproteins) [28]. Proteins are present in the primary cell wall, as a structural component, locking the shape of the cell when elongation is complete [29].

As can be seen in Figure 1, proteins experienced a considerable increase in both MeJ and nano-MeJ-treated grapes in 2019 and 2021, but no differences were shown with respect to control grapes in 2020. Other authors have also shown similar findings, such as Paladines-Quezada et al. [19,20] in two studies carried out during two consecutive years in Monastrell grapes treated with MeJ, which detected an increment in proteins that was more significant in some seasons than in others. With respect to the results shown in 2020, this different behaviour with respect to the two other seasons could be explained by the fact that 2020 was the year in which the grapes were harvested at their ripest moment. This fact could suggest that the higher the degree of maturation, the less the synthesis of proteins in the cell wall is affected by the treatments. This point would be supported by the results found by different authors. Hernández-Hierro et al. [30] observed an increase in proteins during ripening, but this increment seemed to be stagnant in overripe Tempranillo grapes. Ortega-Regules et al. [5] found a positive correlation between °Brix and proteins in a study carried out on premium varieties (Monastrell, Syrah, Merlot and Cabernet) and Huang et al. [31] showed how structural proteins increased with the degree of ripening in grapes (Golden Muscat). Finally, Nunan et al. [32] reported a large change in the amino acid profile of berry cell walls and that proteins (mainly extensins) were deposited in the walls as the berries developed. Regarding other cultivars, MeJ treatment results in an accumulation of proteins (expansin) in the petals leading to cell wall loosening [33].

Therefore, in view of the results shown, it is reasonable to believe that the higher protein content in the skins of treated grapes compared to control grapes could represent a strong structural rigidity, hindering the degradation of CWM from these grape skins.

#### 2.3.2. Total Phenolic Compounds (TPC)

Phenolics are present in the cell wall forming phenolic cross-links with other cell wall components such as lignins, extensins, glucuronoarabinoxylans and side chains of rhamnogalacturonan-I. Such phenolic cross-linking modifies wall strength, cell expansion and pathogen resistance [34]. As can be seen in Figure 1, in general, the results shown in treated grapes with TPC were not different from control grapes. During the three years studied, a remarkable decrease of 31% in TPC was observed when grapes were treated with nano-MeJ in the 2020 season. In contrast, Paladines-Quezada et al. [19,20] showed increases in TPC in Monastrell grapes treated with MeJ in several seasons, although these results were different depending on the year studied.

It is important to note that the degree of ripening of the grapes can affect the TPC content in the cell walls. This fact was detected by different researchers, such as Ortega-Regules et al. [5] in Monastrell grapes, with a small decrease throughout maturing or by Hernández-Hierro et al. [30] who found first an increase and then a slight decrease during ripening of Tempranillo grapes.

#### 2.3.3. Carbohydrate Composition (Uronic Acids, Cellulosic Glucose, and Non-Cellulosic Glucose)

Type I primary cell wall is mainly composed of a network of cellulose microfibrils which is interwoven by hemicellulose forming a cellulose-hemicellulose framework. This is embedded in a matrix of pectin polysaccharides, mainly polygalacturonic acid and rhamnogalacturonan I [29].

As can be seen in Figure 1, uronic acids (UAs), as a measure of pectins, and glucose (cellulosic glucose plus non-cellulosic glucose) were the main components found in our results, with the amount of cellulosic glucose being similar to that shown for uronic acids. Nunan et al. [32] stated that uronic acids and cellulosic glucose were the main components of the grape cell wall.

With regard to cellulosic glucose (cellulose), we can observe that in all treatments, this compound decreased, in contrast to non-cellulosic glucose (hemicellulose), which increased. An increment in hemicellulose, which acts by fixing the cellulose microfibrils [27], would indicate a reinforcement of the cell wall with the treatments. However, other authors, such as Paladines-Quezada et al. [20], in a study carried out in grapes treated with MeJ during two seasons, found a decrease in cellulosic glucose in the first season, but not in the second. Another study by the same author found no effect on cellulosic glucose and a fall in non-cellulosic glucose [19]. Others reports on different cultivars found diverse behaviour: one study in *Bachypodium distachyon* (a model grass) detected an increment in cellulose in response to MeJ treatment [35]. Nevertheless, in a post-harvest treatment and during cold storage of loquat fruits, a decrease in cellulose and hemicellulose was found [36].

As for uronic acids, both treatments increased uronic acids by more than 40% in 2020 and 2021, but not in 2019. During this first season, the highest values were shown in control and nano-MeJ-treated grapes. Since pectins act as a cement between cell wall components [37], the increases observed in treatments during the last two seasons could indicate fortification of the cell wall in these grapes. On the contrary, Paladines-Quezada [19] found that uronic acids were not affected by MeJ treatment over two consecutive seasons in the Monastrell variety, but an increase could be observed in Merlot grapes and a decrease in Cabernet Sauvignon grapes during the same season. The same author in another study found different behaviour depending on the season and variety [20].

#### 2.3.4. Multivariable Factorial Analysis

A multivariable factorial analysis (MFA) was performed to define which factors—season (S), treatment (T), and interaction between them (TxS)—most affected each cell wall compound. Table 3 shows the percentage of variance attributable to each. As can be seen, season affected most of the variables measured in the cell wall: TPC, proteins, cellulosic glucose, and non-cellulosic glucose. Only uronic acids were notably affected by treatment, season and by the interaction between treatment and season. The important influence of season in the different components of the cell wall was also evidenced by other research, such as that of Moreno-Olivares et al. [38], who in a study of cell wall characterisation of new Monastrell hybrid descendants showed differences in most of the components of the cell wall analysed during three consecutive seasons. Likewise, Paladines-Quezada et al. [19], in a report that studied the effect of MeJ and benzothiadiazole on cell wall composition in different grape varieties, showed differences for some cell wall components with respect to the effect of the year.

### 2.4. Wine Anthocyanins and Wine Spectrophotometric Variables

Treatments with MeJ have been reported to induce the phenylpropanoid metabolic pathway, increasing the polyphenol profile. Several reports have demonstrated this, such as a study in plant cell suspension cultures that increased phenolic and flavonoid compounds [39]. In another study on Garnacha, a grape variety characteristically low in phenols, found an increment in treated grapes in total anthocyanins, flavanols and hydroxycinnamic acids [40]. Regarding the Monastrell variety, an increase in anthocyanins, flavonols and skin pro-anthocyanidins was also observed [10]. However, in other studies, although an increase was observed in the grapes, it could not always be seen in the wines [41]. The results of anthocyanin and spectrophotometric variables in the wines are shown in Table 4.

Regarding wine anthocyanins (WA), we can observe an increase in wine from MeJ-treated grapes in 2020 and 2021; however, there was only a slight increase in 2019. These results could be related to the fact that the grapes did not reach full ripening during this year. Ortega-Regules et al. [5] established that an increase in the degree of ripening would be correlated with decreasing amounts of CWM and also with a decrease in the degree of pectin methylation. This would facilitate the release of the compounds found inside the skin cells when the grapes are riper. Ruiz-García et al. [10] found an increment in WA in Monastrell over two vintages. Finally, similar behaviour was observed by Paladines-Quezada et al. [19] in Monastrell wines, with an increase in WA over two seasons, but only in one of them. Regarding the wines from nano-MeJ-treated grapes, the results showed lower or similar values of WA to the wines from control grapes, which may indicate a tendency to decrease with respect to control. Anthocyanins can undergo processes of absorption and release on CWM depending on the temperature and ethanol content. A higher ethanol content in the wine will produce a lower absorption of these compounds [8], so in our case, the average alcoholic degree for each season was 13%, 14.5% and 13.63% in 2019, 2020 and 2021 respectively. These values could imply that in 2019, there was a higher retention of anthocyanins by CWM.

As for IPT, in 2019 and 2020 only the wines from MeJ increased this variable, but in the last season, the treatments did not affect the content of phenolic compounds in their wines. Other authors found a similar trend, such as Ruiz-García et al. [10], with an increment over two seasons in wines from MeJ-treated grapes, and Paladines-Quezada et al. [19], with an increased tendency in wines from the same variety treated with MeJ. Concerning CI, this variable tended to increase in 2020 and 2021 in wines from MeJ-treated grapes. This trend was also found by another author [19], who detected an increase in CI that depended on season and variety. On the contrary, wine from nano-MeJ-treated grapes seemed to decrease it in 2019 and 2020. According to these results, it seems that nano-MeJ treatment had a limited impact on anthocyanins, colour and IPT of the resulting wine. This could be explained by an insufficient dose of MeJ (1 mM) in nano-treatment compared to the conventional treatment (10 mM) or by the overall decrease in dry skin in these grapes, resulting in lower availability of polyphenols and incremented CWM, which could hinder their extraction during winemaking (Table 2).

A multivariable factorial analysis (MFA) was performed to determine which factors, among season and treatment, and the interaction between them, affected each variable. As can be seen (Table 4), the season effect was strongly marked and was the most important factor for the three variables studied. However, treatment effects were 17% for WA, 6% for IPTs and 15% for IC. This limited effect on IPT is in agreement with the fact that phenolic compounds do not only come from the skins but also from the seeds [2].

### 2.5. Pearson Correlation Study

The extraction to a greater or lesser extent of phenolic compounds will depend on the degradation by pectolytic enzymes of grape skins during ripening [30]. To test whether the release of these compounds in wine was affected by the modification of the grape cell wall structure by the treatments, a Pearson correlation study was performed on the three seasons studied. The Pearson correlation measures the strength of the linear relationship between two variables. It has a value between −1 (perfect negative correlation) and 1 (perfect positive correlation).

As can be seen in Figure 2, the IPT of the control wines showed a significant negative correlation with cellulosic glucose (r = −0.73), a trend that changed slightly in treatments for MeJ (r = −0.52) and for nano-MeJ (r = −0.55). It is remarkable the change in trend regarding uronic acids and IPT in both treatments, which went from a negligible negative value (r = −0.16) in control to a significant positive correlation in MeJ (r = 0.75) and nano-MeJ (r = 0.58), which is in accordance with the increase in uronic acids with the treatments (Figure 1). Another remarkable issue was the high negative correlation (r = −0.86) with proteins and the high positive correlation with non-cellulosic glucose (r = 0.69) with nano-MeJ treatment which did not appear in the control or MeJ. Colour intensity displayed a negative correlation with proteins and a positive correlation with hemicellulose, this correlation being maintained in both treatments and being especially important in the case of the nano-MeJ treatment (r = 0.93). In the control, anthocyanins presented a negative correlation with proteins (r = −0.51). This tendency was reinforced in the MeJ treatment, but disappeared in the nano-MeJ one. In addition, MeJ also modified the correlation with TPC (r = 0.56) and hemicellulose (r = 0.79). In the case of nano-MeJ, correlations with uronic acids and cellulosic glucose were also modified, raising the values to r = −0.61 and r = 0.62, respectively.

From these results, we can infer that treatment with MeJ and nano-MeJ not only modify the composition of the cell wall but also influence the synergistic or antagonistic effect that each component may have with the other components on the rigidity of the cell wall. Changes in cell wall strength modify the extractability of the components of the skin cell wall, so factors that strengthen the cell wall will hinder the release of phenolic compounds in the wine, as opposed to those that weaken it.

Another fact to take into account is the different dosage of MeJ in the conventional treatment (10 mM) and in the nanoparticle treatment (1 mM), which could be partly responsible for the differences found in the grapes and wines from the two treatments. Further studies are therefore needed to adjust the dose of MeJ applied on nanoparticles in order to obtain results more similar to the traditional application.

## 3. Materials and Methods

### 3.1. Chemicals

Sodium citrate tribasic dihydrate (Na_3_(C_6_H_5_O_7_)·2H_2_O, ≥99.0% pure) (Na_3_Cit), potassium phosphate dibasic anhydrous (K_2_HPO_4_, ≥99.0% pure), sodium carbonate (Na_2_CO_3_, ≥99.0% pure), calcium nitrate tetrahydrate (Ca(NO_3_)_2_·4H_2_O, ≥99.0% pure), methyl jasmonate (C_13_H_20_O_3_, 95.0%, racemic), Tween 80, pure galacturonic acid, gallic acid, 3,5-dimethylphenol, Bradford reagent, phenol, and Folin–Ciocâlteau phenol reagent were purchased from Sigma-Aldrich (St. Louis, MO, USA). Pure acetone, ethanol 96%, sodium hydroxide 1 N, glacial acetic acid (99%), and sulphuric acid 98% were supplied by Panreac (Barcelona, Spain). An enzymatic analysis kit from TDI (Tecnología Difusion Ibérica S.L., Gavá, Spain) was used for glucose determination and for bovine serum albumin (BSA) fraction V from Roche Diagnostics GmbH (Mannheim, Germany). Ultrapure water came from a Milli-Q system (Millipore Corp., Bedford, MA, USA).

### 3.2. Synthesis of Nanoparticles Doped with MeJ (Nano-MeJ)

For the synthesis of amorphous calcium phosphate nanoparticles, a precipitation method was used at room temperature, based on a previous procedure [23]. Briefly, it consisted of mixing two solutions (1:1 *v*/*v*): (A) Ca(NO_3_)_2_ (0.2 M) and Na_3_Cit (0.2 M) and (B) K_2_HPO_4_ (0.12 M) and Na_2_CO_3_ (0.1 M), under agitation for 5 min. The precipitates were collected and washed with ultrapure water by centrifugation (5000× *g* rpm, 15 min, 18 °C). Subsequently, 200 mg of nanoparticles was vigorously mixed in 10 mL of ultrapure water by vortex and 20 mg of MeJ was added to the nanoparticle suspension. Left for 24 h under agitation at room temperature, nanoparticles doped with MeJ were separated by centrifugation (12,000× *g* rpm, 15 min, 18 °C) and stored at 4 °C.

### 3.3. Vegetal Material and Open Field Treatments

The study was performed over three seasons (2019–2021) in the experimental field El Chaparral located in Cehegin (Murcia, Spain) (latitude 38.11179 and longitude −1.6808), where the climatic conditions are considered semi-arid continental and the soil is loamy sand. The vineyards were *Vitis vinifera* L. Monastrell variety on Richter 110 rootstock, trained in a bilateral cordon system and within-row spacing of 3 × 0.8 m. Three treatments were applied in triplicate at veraison and one week later, following the same protocol described above [24], with 10 vines per replicate: (i) control (water), (ii) 10 mM MeJ and (iii) nano-MeJ at 3.6 g L^−1^ (equivalent to 1 mM in MeJ). When the grapes reached technological maturity (sugar/acidity ratio), a manual harvest was carried out. Subsequently, the grapes were transported in 15 kg boxes to the winery located in the Estación Enológica (Jumilla, Spain).

### 3.4. Physicochemical Characterisation of Grapes

A sample of 300 g was collected from each replicate. The grapes were then crushed with a GT 550 blender (Robot coupe, Montceau-en-Bourgogne, France) and centrifuged for 15 min at 5000 rpm (Meditronic, J.P. Selecta, Barcelona, Spain). In each sample, the following variables were determined: total soluble solids (°Brix) using an Atago RX-5000X refractometer (Atago Co., Ltd., Tokyo, Japan), pH and titratable acidity with a Schott, alpha plus TA20 (SCHOTT-GERÄTE GmbH, Mainz, Germany) using a glass electrode (Xylem analytics Germany GmbH, Weilheim, Germany).

### 3.5. Isolation of Cell Wall Material (CWM)

The cell wall was deconstructed according to Apolinar-Valiente et al. [4] and Paladines-Quezada et al. [19]. At the time of harvest, grape samples (100 g for each treatment) were frozen at −20 °C. Subsequently, the skins were peeled with a scalpel and stored at −20 °C until isolation of the cell wall material. The frozen skins were lyophilized with a Cryodos 50 (IMA-TELSTAR, Terrassa, Spain) and pulverised in a ball mill (Vibratory Ball Mill Pulverisette 0, Cryo-box, FRITSCH, Idar-Oberstein, Germany). The pulverised grape skins were suspended in 50 mL of boiling water for 5 min to inactivate the enzymes, then centrifuged and the supernatant was removed. The sample was washed for 30 min at 40 °C with 70% ethanol, centrifuged and the supernatant removed. This process was repeated until all soluble sugars were eliminated from the supernatant and verified by the Dubois method [42]. The alcohol-insoluble solids were washed twice with 96% ethanol and once with acetone. Finally, samples were dried overnight under a stream of air at 20 °C.

### 3.6. Analysis of the Composition of Grape Skin Cell Wall

The cell wall structure is complex. In short, it is composed of a network of structural proteins and polysaccharides, the latter containing hemicellulose, pectins and cellulose microfibrils [43]. In addition, phenolic compounds are bound to proteins and polysaccharides [44]. To obtain an approximation of the cell wall structure, the following variables were analysed on isolated CWM (10 mg for each assay) and in quadruplicate for each treatment: proteins, total phenolic compounds, uronic acids, total glucose, cellulosic glucose and non-cellulosic glucose.

#### 3.6.1. Proteins and Total Phenolic Compounds

The content of proteins and total phenolic compounds was determined in the supernatant after extraction of CWM with 1 M NaOH (100 °C, 10 min). Proteins were analysed by the colorimetric Coomassie Brilliant Blue assay [45], bovine serum albumin (BSA) fraction V was used to perform the calibration curve, and the results were expressed as mg BSA g^−1^ of cell wall. The determination of total phenolic compounds (TPC) was performed by the colorimetric Folin-Ciocâlteau reagent test [46]. The calibration curve was carried out using a solution of gallic acid and the results were expressed as mg gallic acid g^−1^ cell wall.

#### 3.6.2. Uronic Acids and Glucose

Uronic acids, as a measure of pectins, and total glucose were determined after pre-hydrolysis (30 °C, 1 h) with aqueous 72% sulfuric acid, followed by hydrolysis with sulphuric acid 1 M (100 °C, 3 h). Uronic acids were analysed by a colorimetric 3.5-dimethylphenol assay [47]. Pure galacturonic acid was used as standard for the calibration curve.

For the determination of non-cellulosic glucose (hemicellulose), only hydrolysis with 1 M sulphuric acid (100 °C, 3 h) was performed. The determination of both glucoses was made with a kit for glucose enzymatic analysis from TDI using a Miura 200 (Rome, Italy). The cellulosic glucose (cellulose) was obtained by difference between total glucose and non-cellulosic glucose content.

### 3.7. Vinifications

The grapes were destemmed, crushed and sulphited (0.08 g SO_2_/kg). The must acidity was then adjusted to 5.5 g L^−1^ with tartaric acid and inoculated with yeast at 0.25 g L^−1^ (Zymaflore FX10, Laffort, Bordeaux, France). All vinifications, three replicates for each treatment, were made in 50 L stainless steel tanks. Alcoholic fermentation was carried out at a temperature of 23 ± 2 °C for 14 days, punching the cap down daily. Once alcoholic fermentation was completed, each replicate was pressed, and the free-run and pressed wine were collected in 50 L stainless steel tanks. After racking twice over two days, each replicate was stored in a bag in box.

### 3.8. Spectrophotometric Variables in Wines

All variables were measured on wine samples at the end of alcoholic fermentation in triplicate. Colour intensity (CI) and total polyphenols (IPT) were analysed using a Shimadzu UV/visible spectrophotometer 1600PC (Shimadzu, Duisburg, Germany). The CI was calculated as the sum of the absorbance at 620 (blue), 520 (red), and 420 (yellow) nm in undiluted wine [48] and the IPT were analysed by measuring absorbance at 280 nm [49]. Total wine anthocyanins (WA) were determined by the colorimetric method based on the Puissant–Léon technique [50] with an automatic analyser CETLAB 600 (Microdom, Taverny, France).

### 3.9. Statistical Analysis

Significant differences between treatments for each variable were assessed by analysis of variance (ANOVA) and multifactorial analysis of variance (MANOVA) using RStudio 3.6.2 (Boston, MA, USA). A least significant difference (LSD) test was used to compare the means (*p* < 0.05) when the ANOVA test was significant.

In addition, Pearson correlation coefficients of IPT, colour intensity, anthocyanins, uronic acids, total phenolic compounds, proteins, cellulosic glucose and non-celullosic glucose were calculated using the statistical package Statgraphics Centurion 18.

## 4. Conclusions

From the results obtained, it can be deduced that the foliar application of MeJ increased the proportion of dry skin in Monastrell grapes, which allowed a greater availability of compounds of interest contained in it (such as colour pigments and phenolic compounds). MeJ and Nano-MeJ increased the CWM isolated from the skins, which suggested that the treatments tend to produce a thicker cell wall that may hinder the extraction of compounds of interest in the wine. In addition, both treatments modified the composition of the skin cell wall: an increase in uronic acids was detected when the grapes were properly ripened.

Regarding phenolic and spectrophotometric variables of the wines, only MeJ was able to increase the anthocyanins in the wines when the grapes reached adequate maturity, producing a tendency towards an increase in CI. On the contrary, nano-MeJ treatment had a negative impact on CI, perhaps due to the reduction of dry skin in these grapes, which resulted in lower availability of polyphenols and an increase in CWM that could hinder their extraction during winemaking. Furthermore, IPT was not affected by either treatment, despite the correlation with the components of the cell wall that changed in both treated grapes.

Although some variables were modified by the treatments, the results showed that MeJ affected grapes and wines to a greater extent than nano-MeJ, which may be due to the lower doses in the latter treatment. Despite these results, season was the most important factor that affected both the wine spectrophotometric variables and the composition of the Monastrell cell wall. Therefore, further studies should be carried out to establish an optimisation of the application of MeJ in nanoparticle form.

## Figures and Tables

**Figure 1 molecules-28-01478-f001:**
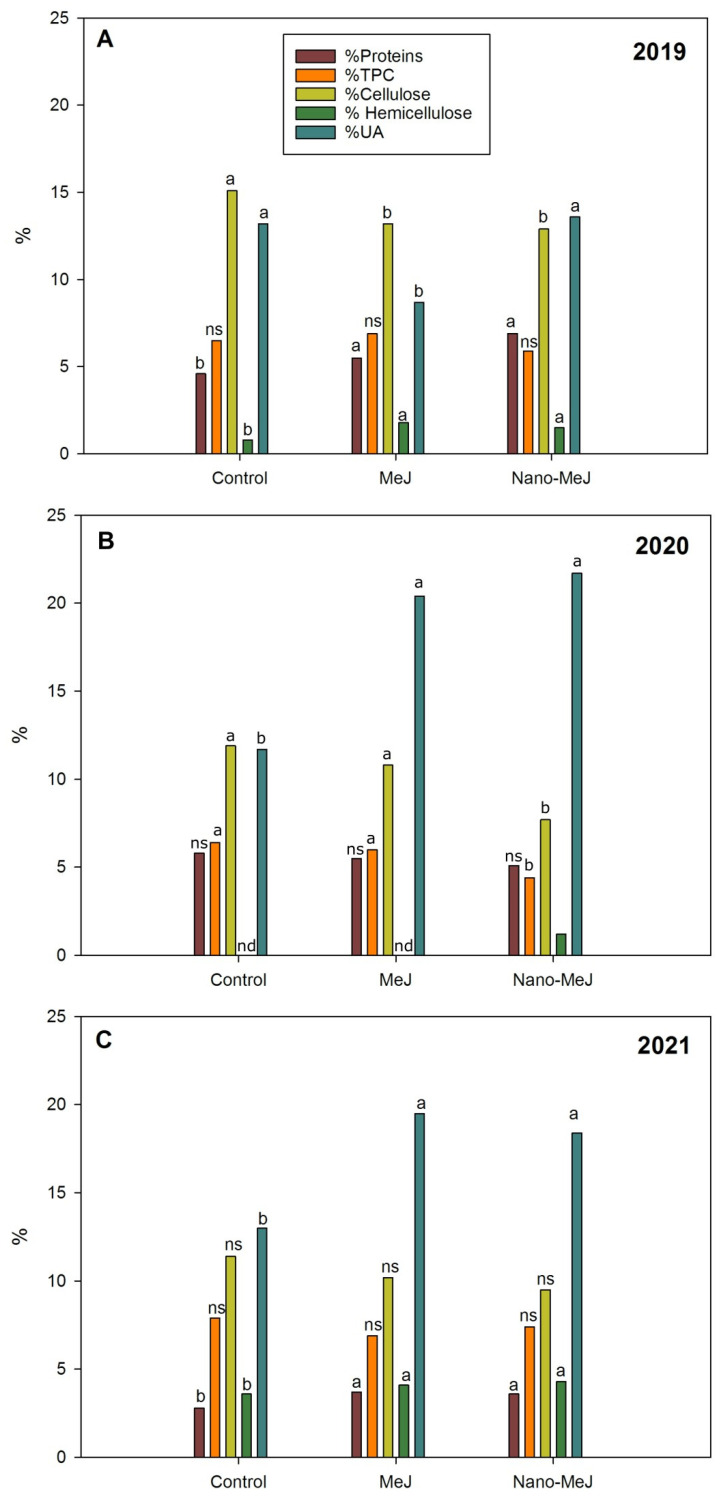
Skin cell wall composition in Monastrell grapes untreated and treated with MeJ and nano-MeJ during three years ((**A**) 2019, (**B**) 2020 and (**C**) 2021). Different letters with the same bar colour indicate significant differences according to LDS test. (ns; not significant).

**Figure 2 molecules-28-01478-f002:**
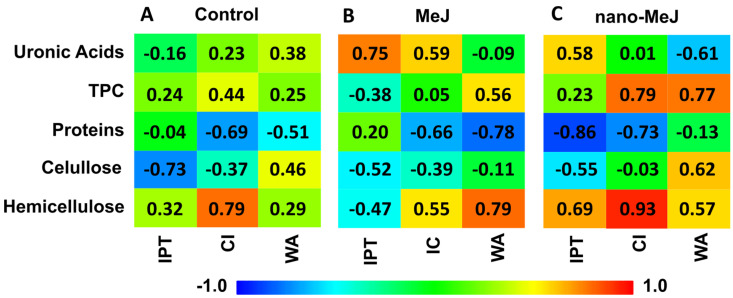
Pearson correlation coefficients between cell wall components and wine spectrophotometric characteristics for control, MeJ and nano-MeJ treatment.

**Table 1 molecules-28-01478-t001:** Physicochemical characteristics of control and treated grapes (MeJ and nano-MeJ) and harvest date during three years.

			Weight (g) (100 Berries)	Soluble Solids (°Brix)	TA (g of Tartaric Acid L^−1^)	
		Harvest Date				pH
**2019**	**Control**	23 September 2019	154.98 ± 5.66	22.07 ± 0.58	4.17 ± 0.24 b	3.74 ± 0.02
	**MeJ**	23 September 2019	154.64 ± 10.99	22.27 ± 0.55	4.74 ± 0.17 a	3.74 ± 0.05
	**Nano-MeJ**	23 September 2019	158.91 ± 5.24	22.72 ± 0.56	4.68 ± 0.01 a	3.78 ± 0.02
	** *p* ** **-value**		ns	ns	*	ns
**2020**	**Control**	29 September 2020	162.79 ± 3.69	25.75 ± 0.33	2.87 ± 0.00 b	4.14 ± 0.06
	**MeJ**	29 September 2020	165.29 ± 9.52	24.92 ± 0.38	3.08 ± 0.08 a	4.13 ± 0.04
	**Nano-MeJ**	29 September 2020	170.87 ± 4.83	25.35 ± 0.35	2.87 ± 0.13 b	4.16 ± 0.03
	** *p* ** **-value**		ns	ns	*	ns
**2021**	**Control**	4 October 2021	143.70 ± 11.00 b	23.27 ± 0.60	2.75 ± 0.20 b	4.02 ± 0.03
	**MeJ**	4 October 2021	165.84 ± 7.98 a	23.83 ± 0.25	3.38 ± 0.16 a	4.02 ± 0.04
	**Nano-MeJ**	4 October 2021	168.67 ± 0.59 a	23.50 ± 0.17	2.72 ± 0.25 b	4.04 ± 0.03
	** *p* ** **-value**		*	ns	*	ns

Different letters in the same row indicate significant differences according to LDS test (statistically significant at * *p* ≤ 0.05). ns, not significant.

**Table 2 molecules-28-01478-t002:** Percentage of dry skin in grapes and percentage of cell wall in the dry skin of different treatments during three years.

		% Dry Skin	% CWM
		In Grape	(In Dry Skin)
**2019**	**Control**	3.02 b	25.66 b
	**MeJ**	3.17 a	28.55 a
	**Nano-MeJ**	2.88 c	28.30 a
	** *p* ** **-value**	***	***
**2020**	**Control**	3.26	24.94 b
	**MeJ**	3.32	24.79 b
	**Nano-MeJ**	2.98	27.76 a
	** *p* ** **-value**	ns	***
**2021**	**Control**	3.08 b	22.62 c
	**MeJ**	3.33 a	24.66 a
	**Nano-MeJ**	3.15 b	23.54 b
	** *p* ** **-value**	*	***

CWM, cell wall material. Different letters within the same year represent significant differences according to an LDS test (*** *p* ≤ 0.001, * *p* ≤ 0.05, ns, not significant).

**Table 3 molecules-28-01478-t003:** Multivariable factorial analysis of cell wall components with treatment, season and treatment–season interaction.

	T (%)	S (%)	TxS (%)	Error (%)
**Uronic Acids**	24 ***	37 ***	31 ***	7
**TPC**	15 **	40 ***	13 *	32
**Proteins**	7 ***	69 ***	17 ***	8
**Cel-glu**	25 ***	57 ***	5 ns	13
**Non-cel-glu**	4 ***	88 ***	2 *	5

TPC: total phenolic compounds, Cel-glu: cellulosic glucose, non-cel-glu: non-cellulosic glucose. Different letters within the same column represent significant differences according to an LDS test (*** *p* ≤ 0.001, ** *p* ≤ 0.01, * *p* ≤ 0.05).

**Table 4 molecules-28-01478-t004:** Wine spectrophotometric variables from control and different treatments (MeJ and nano-MeJ).

		WA	IPT	CI
		(mg L^−1^)		
**2019**	Control	529.00 ± 38.97	39.11 ± 1.05 b	14.01 ± 1.03
	MeJ	551.00 ± 24.00	41.85 ± 0.97 a	14.01 ± 0.81
	Nano-MeJ	520.33 ± 44.00	41.32 ± 2.71 a	12.94 ± 0.67
	*p*-value	ns	*	ns
**2020**	Control	572.33 ± 29.78 b	47.88 ± 1.76 b	14.14 ± 0.85
	MeJ	682.67 ± 49.51 a	53.75 ± 0.96 a	15.55 ± 3.52
	Nano-MeJ	527.67 ± 44.40 b	44.89 ± 2.91 b	12.43 ± 0.90
	*p*-value	*	**	ns
**2021**	Control	621.67 ± 56.72 b	47.24 ± 2.45	16.37 ± 1.47
	MeJ	739.00 ± 54.15 a	45.52 ± 3.81	18.29 ± 0.88
	Nano-MeJ	635.33 ± 20.03 b	47.81 ± 0.71	16.56 ± 0.48
	*p*-value	*	ns	ns
	T (%)	17 ***	6 **	15 **
**Multifactorial**	S (%)	63 ***	61 ***	55 ***
**analysis**	TxS (%)	8 **	21 ***	5 ns
	Error (%)	11	12	25

WA, wine anthocyanins; CI, colour intensity Different letters in the same row indicate significant differences according to LSD test (*** *p* ≤ 0.001, ** *p* ≤ 0.01, * *p* ≤ 0.05, ns, not significant).

## Data Availability

Data are contained within the article.

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
