# Peer review of "Methyl Jasmonate and Nanoparticles Doped with Methyl Jasmonate affect the Cell Wall Composition of Monastrell Grape Skins"

_molecules, 2023, doi:10.3390/molecules28031478_

Round 1

Reviewer 1 Report

The aim of this research was to investigate the composition of the cell wall of Monastrell grapes treated with MeJ and nano-MeJ and its relationship with the chromatic and phenolic characteristics of the wines, during three seasons.

The focus of the study is interesting for winemakers, and wine consumers.

However, several aspects of the paper should be revised.

In first, the dose of MeJ was different in both treatments. I think that the authors should be explain that, because it is impossible relate the results with the application form of MeJ without refer these results to the doses of product.

Another aspect to review is the data considered for the wines. The evaluated parameters are scarce, and at least the hue could be included, to have a better description of the color.

Finally, in the Conclusions were included comments about the incidence of the doses of MeJ. I think that these references should be increased in the Discussion of the results.

Author Response

it better understood by readers.

Regarding your comments:

In first, the dose of MeJ was different in both treatments. I think that the authors should be explain that, because it is impossible relate the results with the application form of MeJ without refer these results to the doses of product.

It is possible that the introduction did not explain very clearly why we decided to do the study of the effect of nano-MeJ, the fact that MeJ concentration are different is because the aim is to use nanotechnology to have to use lower concentrations, lower cost and in a more sustainable way. We have introduced an explanation to make it clearer:

MeJ is an expensive compound with high volatility, which is why it is necessary to optimize the application method in order to reduce the dose and avoid losses due to volatization or degradation. This would help to reduce costs and achieve a more sustainable agriculture. That is the reason that led us to investigate how grapes and wines are affected after the application of MeJ supported on nanoparticles, using lower doses than the treatment with MeJ applied in a conventional way.

Another aspect to review is the data considered for the wines. The evaluated parameters are scarce, and at least the hue could be included, to have a better description of the color.

We have used the parameters that we consider most important according to our experience. Specifically, in the case of new wines with an identical pH, we do not consider the hue variable to be important, as it does not change substantially in the case of wines from the same variety and with identical maceration times.

Finally, in the Conclusions were included comments about the incidence of the doses of MeJ. I think that these references should be increased in the Discussion of the results.

We included in 2.4 of Results and discussion the reference about de dose of MeJ:

 According to these results, it seems that nano-MeJ treatment had a limited impact on anthocyanins, colour and IPT of the resulting wine. This could be explained by an insufficient dose of MeJ (1mM) in nano treatment compared to the conventional treatment (10 mM) or by the overall decrease of dry skin in these grapes which resulted in a lower availability of polyphenols and incremented CWM that could hinder their extraction during winemaking (Table 2).

However, it may be necessary to emphasise this fact and also that the different behaviour found in some of the cell wall components could also be due to the difference in application rate. So we have added the following sentence at the end part of Results and discussion:

Another fact to take into account is the different dosage of MeJ in the conventional treatment (10mM) and in the nano-particle treatment (1mM), which could be partly responsible for the differences found in the grapes and wines from the two treatments. Further studies are therefore needed to adjust the dose of MeJ applied on nano-particles in order to obtain results more similar to the traditional application.

Reviewer 2 Report

Please see my attached review

Author Response

We appreciate your comments as they help us to improve the article and make it better understood by readers.

Regarding your comments:

ABSTRACT

Line 15-16. Reword first sentence. You are saying that the composition = contents. Of course!

It is true that we have not expressed properly. We have rewritten the sentence:

The structural composition of the cell wall of grape skins is related to the cell wall integrity and  the subsequent extraction of the different compounds that are contained inside vacuoles and also the cell wall breakdown products.

Line 16. Define MeJ here as was done in l. 19.

We have done this

Line 19 . Just use MeJ here and define in l. 16.

We have done this

Line 21 The results indicated that Both

We have rewritten the sentence:

Both treatments tended to increase cell wall material (CWM)……

Line 23, 24 Avoid use of 'significant(ly)'--it is a scientific redundancy. Just say they were different. I suggest deletion of the word “significant(ly)” throughout the manuscript, as well as “statistically significant”.

We have removed “significant(ly)” where it was redundant

INTRODUCTION

Line 41 …braking down breakdown of the…

We have done this

Line 49, 50 Botrytis (uppercase); downy mildew need not be in italics.

We have done this in the manuscript

RESULTS & DISCUSSION

Line  75. “Total acidity”--more correctly, "titratable acidity"; here and elsewhere in

the manuscript, e.g. l. 92. Or just abbreviate to 'TA' after first mention and

definition.

We have done this in the manuscript

Line 85. Suggest use of "variable(s)" here and elsewhere instead of "parameters".

We have done this in the manuscript

Line 92 …action regarding response by titratable total acidity…

We have done this in the manuscript

Line 94 Table 1: Please revise table title so that this table stands on its own—e.g.

location, variety, years, etc.

We have modified the title of this Table.

Line 119 Table 2: Please revise table title so that this table stands on its own.

We have modified the title of this Table.

Line 129 Figure 1: Please revise figure title so that this figure stands on its own

We have modify the title of this Figure.

We have detected missing statistical letters on fig 1.B, now it is correct.

Line 138 …no statistical differences…

We have corrected this in the manuscript

Line 155 Is this correct? Do you mean petioles?

No, petals is correct

Line 166 … not statistically significant different in…

We have corrected this in the manuscript

LINE 195. Italicize species name.

We have corrected this in the manuscript

Line 220 …showed significant differences…

We have corrected this in the manuscript

Line 222. Table 3: Please revise table title so that this table stands on its own.

We have modify the title of this Table.

Line 237 Table 4: Please revise table title so that this table stands on its own.

We have modify the title of this Table.

Line 240 … observe a significant an increase…

We have corrected this in the manuscript

Line 241 … however the results showed there was only

We have corrected this in the manuscript

Line 243 …established… (‘e’ missing)

We have corrected this in the manuscript

Line 249 … only significantly in one…

We have corrected this in the manuscript

Line 252 …control, although not significantly.

We have corrected this in the manuscript

Line 262-266

…increased but not statistically significant tendency in wines from the

same variety treated with MeJ.

We have corrected this in the manuscript

Concerning CI, this parameter variable increased tended to increase in

2020 and 2021 in wines from MeJ treated grapes but not significantly, this

trend was also found by another author [19] who detected an increment increase

in CI that was significant dependeding

We have corrected this in the manuscript

Line  288

Fig. 1: Is this a figure or just a table with different colored cells? I think it is a table. Whether figure or table, please revise its title so that this table stands on its own. Note also that you already have a Fig. 1.

Because the code color that is on its foot we prefer to define as a figure instead of a table. However, it is true that a figure 1 already exists, it is a numbering error, which we have corrected in the manuscript.

Materials & Methods

Line 320, 417,    Delete “USA” here and elsewhere.

We have corrected this in the manuscript

Line 358      “harvest” (‘t’ is missing)

We have corrected this in the manuscript

Conclusions

Line   429       …wall, an significant increase…

… producing an increment a tendency toward an increase in CI but it was not statistically significant.

We have corrected this in the manuscript

Line  437      Delete “Pearson”.

We have corrected this in the manuscript

Round 2

Reviewer 1 Report

The paper was revised by the authors. The new paragraphs or that modified clarified the questions realized. I think that the paper can be publish in the present form.